# PROSHADE Protocol: Designing and Evaluating a Decision Aid for Promoting Shared Decision Making in Opportunistic Screening for Prostate Cancer: A Mix-Method Study

**DOI:** 10.3390/ijerph19158904

**Published:** 2022-07-22

**Authors:** Blanca Lumbreras, Lucy Anne Parker, Pablo Alonso-Coello, Javier Mira-Bernabeu, Luis Gómez-Pérez, Juan Pablo Caballero-Romeu, Salvador Pertusa-Martínez, Ana Cebrián-Cuenca, Irene Moral-Peláez, Maite López-Garrigós, Carlos Canelo-Aybar, Elena Ronda, Mercedes Guilabert, Antonio Prieto-González, Ildefonso Hernández-Aguado

**Affiliations:** 1Department of Public Health, History of Science and Gynecology, Miguel Hernandez University, 03550 San Juan de Alicante, Spain; lparker@umh.es (L.A.P.); ihernandez@umh.es (I.H.-A.); 2CIBER of Epidemiology and Public Health, CIBERESP, 28029 Madrid, Spain; 3Iberoamerican Cochrane Centre, Department of Clinical Epidemiology and Public Health, Biomedical Research Institute Sant Pau, 08001 Barcelona, Spain; palonso@santpau.cat (P.A.-C.); carlos.canelo.ay@gmail.com (C.C.-A.); 4Department of Preventive Medicine, Hospital Universitario de San Juan, 03550 San Juan de Alicante, Spain; havimira@hotmail.com; 5Department of Urology, Hospital Universitario de San Juan, 03550 San Juan de Alicante, Spain; luisgope@gmail.com; 6Department of Urology, Hospital General Universitario de Alicante, 03010 Alicante, Spain; juanpablocaballero@gmail.com; 7Cabo Huertas Healthcare Centre, 03540 Alicante, Spain; salvador.pertusa@gmail.com; 8Cartagena Casco Healthcare Centre, 30201 Cartagena, Spain; anicebrian@gmail.com; 9Unidad de Investigación, Equipo de Atención Primaria Sardenya, Instituto de Investigación Biomédica Sant Pau, 08001 Barcelona, Spain; imoral@eapsardenya.cat; 10Clinical Laboratory, Hospital Universitario de San Juan, 03550 San Juan de Alicante, Spain; maitelopezgarrigos@gmail.com; 11Public Health Research Group, Alicante University, 03690 San Vicente del Raspeig, Spain; elena.ronda@ua.es; 12Department of Health Psychology, Miguel Hernandez University, 03202 Elche, Spain; mguilabert@umh.es; 13Cancer Prostate Association (ANCAP), 30012 Murcia, Spain; antonioprietogonzalez@hotmail.com

**Keywords:** prostate cancer screening, prostate specific antigen (PSA), shared decision making (SDM)

## Abstract

**Background:** Opportunistic prostate-specific antigen (PSA) screening may reduce prostate cancer mortality risk but is associated with false positive results, biopsy complications and overdiagnosis. Although different organisations have emphasised the importance of shared decision making (SDM) to assist men in deciding whether to undergo prostate cancer screening, recent evaluations show that the available decision aids fail to facilitate SDM, mainly because they do not consider the patients’ perspective in their design. We aim to systematically develop and test a patient decision aid to promote SDM in prostate cancer screening, following the Knowledge to Action framework. **Methods:** (1) Feasibility study: a quantitative survey evaluating the population and clinician (urologists and general practitioners) knowledge of the benefits and risks derived from PSA determination and the awareness of the available recommendations. Focus groups to explore the challenges patients and clinicians face when discussing prostate cancer screening, the relevance of a decision aid and how best to integrate it into practice. (2) Patient decision aid development: Based on this data, an evidence-based multicomponent SDM patient decision aid will be developed. (3) User-testing: an assessment of the prototype of the initial patient decision aid through a user-testing design based on mix-methods (questionnaire and semi-structured review). The decision aid will be refined through several iterative cycles of feedback and redesign. (4) Validation: an evaluation of the patient decision aid through a cluster-randomised controlled trial. **Discussion:** The designed patient decision aid will provide balanced information on screening benefits and risks and should help patients to consider their personal preferences and to take a more active role in decision making. **Conclusions:** The well-designed patient decision aid (PDA) will provide balanced information on screening benefits and risks and help patients consider their personal preferences.

## 1. Background

Shared decision making (SDM) has been defined as the key to successful patient-centred care [1], and consequently, researchers, clinicians, patients and health policy representatives have made a considerable effort to implement SDM in clinical practice [2]. Current clinical practice guidelines for the early detection of prostate cancer recommend for clinical decision making a personalised prostate-specific antigen (PSA)-based management to improve the risk–benefit ratio of the screening strategy. Some important critical issues regarding the PSA determination (i.e., the poor harmonisation of the assays) in the clinical framework are, however, still neglected in the current guidelines, and a major focus of recommendations on those aspects would be needed to improve their effectiveness [3,4]. In addition, evidence suggests that clinicians have traditionally underestimated the adverse impact of PSA [5], and consequently, it is rarely explained to patients [6], although several studies show that most patients would like to be informed [7]. According to previous data, fewer than 30% of men discuss PSA screening with their clinicians, and these dialogues do not promote SDM [8], although it is critical to assist men in making informed decisions in opportunistic screening.

The decision to undergo PSA testing for prostate cancer is a difficult one for men, and patient decision aids have been proposed to promote SDM. Patient decision aids (PDA) are tools that help patients learn about a condition and review the possible benefits, harms and scientific uncertainties about potential options. They are particularly useful when the efficacy and outcomes are unclear, as well as when the outcomes are clear but the trade-off between benefits and risks requires subjective judgment, as in PC screening, where most men overestimate the benefits of PC screening and are unaware of the limitations. These issues, as well as difficult concepts such as overdiagnosis and overtreatment, make decision aids especially useful in approaching the clinician–patient discussion in PC screening [9]. In 2010, the Spanish Ministry of Economy and Competitiveness evaluated PDAs for patients with PC to help them to decide on their treatment and management [10]. However, there are no evaluations in PDAs for PC screening.

Several randomised clinical trials have evaluated PDAs for PC screening, mainly among primary care patients, with contradictory results. According to a recent meta-analysis [11], the available PDAs face both design and implementation challenges, and although they improve men’s knowledge regarding PSA benefits and risks, they fail to promote SDM. Hence, how to design and implement PDAs in PC screening represents a large gap in the existing literature. The meta-analysis identified the lack of consideration of the patients’ perspective as the main concern in the design and implementation of these previous PDAs, resulting in a poor analysis of the best strategy to help patients to evaluate which benefits and harms are most important to them. There are many factors that may determine whether men receive screening, including their personal preferences and factors related to their clinician. Recent research has highlighted that men do not have uniform reactions to PC screening information: some prefer an active approach and want to receive PSA screening after being informed, while others prefer not to receive it [12]. There are also some PC screening determinants which can influence the patient’s decision such as education, age and prior PSA testing, but these factors have not been adequately addressed in previous decision-aids. Some studies have proposed the use of a well-calibrated risk prediction model to define PSA thresholds for identifying or excluding advanced PC as an aid to personalise the management of the diagnostic workup. The proposed PSA thresholds, being associated with the related predictive values, may allow for an individualised approach to the diagnostic workup, assisting patients in making an informed decision [13].

Barriers also exist at multiple levels in the healthcare system [14]. Some clinicians do not agree with the information included in the tools. The pressure to see a larger volume of patients in a shorter period of time [15] is also a known barrier, together with a lack of experience in SDM [16]. However, previous PDAs did not consider the patients and clinicians’ characteristics, preferences and attitudes. In addition, patients should receive information contextualised to their particular setting in order to help them in their informed decision. Information regarding test properties, such as the likelihood of having a false- positive result or overdiagnosis, are not frequently explained to patients [17]. Moreover, most of the available data derives from clinical trials with high-risk populations, which differ quite significantly from the unselected population undergoing opportunistic screening in clinical practice. Our group has previously evaluated those factors associated with the presence of false-positive and -negative results in PSA determinations carried out in clinical practice through the inclusion of 572 men with a negative PSA result and 1081 mean with a positive result from 20 primary health centres in the Valencian Community who have been followed up for 2 years [18]. The information of this previous study about the probability of having a false positive result will provide patients with information applicable to their setting.

Therefore, this study seeks to overcome the fact that available PDAs have shown no effect on screening discussion, patient’s decision satisfaction or actual screening. The main result will be a PDA that will not only provide patients with information but will also specifically target the promotion of SDM. This PDA will be evaluated through a trial to mimic the conditions in primary care, where the distribution of a decision aid has been recommended.

The overall aim of this project is to systematically develop and test a PDA to promote SDM in PC screening, following the Knowledge to Action (KTA) framework [19].

## 2. Materials and Methods

This study will be guided by the KTA framework [19] to develop, test and validate a PDA to promote SDM in PC screening. The KTA framework is adequate for this study because it includes the integration of different sources of evidence together with researchers and knowledge-users to improve the decision process in PC screening and to provide more effective health services.

### 2.1. Study Overview

The study detailed in this protocol paper consists of four phases: feasibility testing (Phase 1); PDA development (Phase 2); usability testing (Phase 3); and evaluation in a cluster random controlled trial study (Phase 4) (Figure 1).

#### 2.1.1. Phase 1: Feasibility Test

In this phase, we will assess the feasibility of the PDA. First, we will evaluate the knowledge among the target population and clinicians (urologists and general practitioners) about the benefits and risks of opportunistic screening for PC and the available recommendations through a quantitative survey. Second, through focus groups, we will explore the challenges men and clinicians face when addressing the benefit and risk of PC screening, including the core information both groups need to establish an informed discussion. We will also analyse the patients’ and clinicians’ attitudes, reactions and preferences regarding different formats of a PDA and establish the preferential criteria for its implementation in practice.

##### Participants

Knowledge surveys

For the population’ s knowledge evaluation, we will survey the population belonging to the Valencian Community, Spain (according to the Spanish National Statistics Institute, in 2021, there were 1,389,725 men > 40 years). We will include men > 40 years old living in the Valencian Community, and men who have a PC diagnosis will be excluded. We estimate that, for a precision of 5% with a 95% Confidence Interval (CI), at least 1067 men would be required on a conservative estimate that 50% of the men could be unaware of the PSA benefits and risks. This precision will allow for analysis by subgroups. The sampling will be conducted by geographic area (the three provinces of the Valencian Community) and population habitat (rural and urban) and will be stratified by age (40–50; 50–70; >70).

For the clinician (urologist and general practitioners) knowledge surveys, we will select clinicians through their respective scientific societies. According to a previous survey [20], 56.1% of the general practitioners and 64.3% of the urologists discussed the impact of PSA on mortality with patient; for a precision of 5% (95% CI), 369 general practitioners and 345 urologists will be surveyed after being randomly selected from the list obtained from scientific societies.

2.Focus groups with users

To explore clinicians’ perceptions, we will include a purposive sample of urologists and general practitioners working in two Health Departments in the Valencian Community (Health Department Alicante-General Hospital, 255,439 habitants; and Health Department Alicante-S. Joan d’Alacant, 233,115 habitants).

To explore patients’ perceptions, eligible patients (men > 40 years who do not have a PC diagnosis and are willing to participate in focus groups, with a varied profile of age and educational level) will first be identified by the general practitioners working in the two Health Departments in the Valencian Community. The clinicians will ask them for their permission to be contacted by a member of the research team, who will then offer them to participate in the study.

At least three focus groups of patients and clinicians (8–12 participants in each group) will be made to triangulate the information and obtain cross-validity. We will also carry out one focus groups with 8–12 participants each belonging to the Prostate Cancer Association (ANCAP) in Spain in order to compare the outcomes with those who do not have a PC diagnosis.

##### Data Collection

Knowledge surveys

The general population will be surveyed by phone (through a Computer-assisted Telephone Interviewing -CATI- platform by the random selection of telephone numbers), indicating the purpose of the study and requesting oral informed consent. Clinicians will be contacted and invited to take part through their scientific societies, and we will survey them using a Google questionnaire. Both surveys will be developed after reviewing the available scientific literature and will be validated through the Delphi method with experts (urologists, general practitioners, epidemiologists and psychologists). The final surveys will be piloted before use, and adaptations will be made to improve their clarity.

2.Focus groups with users

Each focus group will be performed according to a previously established topic guide considering the following issues: discussion about the barriers and facilitators to promoting SDM and their relationship with the information patients need; aspects related to the patient’s role (passive or active) in screening decision making. In addition, several procedures for presenting the information, including different formats, will be presented to evaluate the patients and clinicians’ reactions and preferences, according to the Control Preferences Scale [21]. They will also be asked about the ideal setting and moment in the decision-making process to present the decision aid. A trained interviewer will conduct each focus group. All discussions will be audiotaped, and field notes will be kept.

##### Analysis

Knowledge surveys

The demographic data of the surveyed subjects will be coded. The frequency of response will be described in each of the items of the survey, expressed with 95% confidence intervals, and statistically significant differences in selected independent variables will be analysed using the Pearson χ^2^ test for categorical variables and the Mann–Whitney U test for continuous variables, (*p* < 0.05).

2.Focus groups with users

Two researchers will triangulate the information obtained from the different focus groups. First, a careful transcript reading will be carried out, and the text will be split up into meaningful information units. These units will be coded following a mixed strategy (emerging and predefined codes according to the study objectives), and categories will be developed based on grouping codes with the same theme. Finally, the points of agreement and disagreement will be analysed.

#### 2.1.2. Phase 2: Patient Decision Aid Development

Based on the findings from Phase 1 and a review of the available evidence, we will follow the standards of the International Patient Decision Aid Standards Collaboration [22] to develop the evidence-based multicomponent PDA. The International Patient Decision Aids Standards provide explicit guidance on content, the development process and effectiveness.

The PDA will include information about PSA application in the early detection of PC; available recommendations from the European Urologist Society [23]; well-balanced information about population-specific benefits and risks based on individual risk factors and the previous results in this setting [15]; and a discussion of the limitations of the results (false-positive and false-negative results, overdiagnosis). The initial PDA will be in the Spanish language; the patient-directed components will target a Grade 8 literacy level (readability score) [24]. The PDA will be designed in an iterative fashion by study team members, including a graphics designer and a computer programmer to include a combination of numbers, graphics and narratives.

#### 2.1.3. Phase 3: User-Testing

In this phase, we will assess the prototype of the initial PDA through a user-testing design based on mix-methods (a questionnaire and a semi-structured interview). The PDA will be refined through several iterative cycles of feedback and redesign.

##### Participants

As in Phase 1, men > 40 years of varied socio-demographic profiles (described in Phase 1) who do not have a PC diagnosis will be contacted by their general practitioners. We will also include patients from the ANCAP (contacted by the urologist president of the association) and a convenience sample of clinicians (urologists and general practitioners) not involved in the development process for content accuracy. Research has shown that up to 80% of usability issues can be identified through five to eight participants [25]. We anticipate conducting up to three usability cycles of four to five participant dyads in a process of iterative redesign.

##### Data Collection

To user-test, a consultant with experience will administer a questionnaire to determine whether the individual can understand selected information key points. A short semi-structured interview will be conducted to explore the participants’ views.

After each round of testing, the findings will be analysed to overcome identified problems. The process will be repeated until all problems have been satisfactorily resolved. Participants will also be interviewed regarding the degree of satisfaction and the strengths and weaknesses of the PDA. All interviews will be audiotaped.

##### Analysis

The audiotapes will be analysed independently by two researchers. We will categorise and organise the data as the initial thematic analysis, which will allow us to identify recurrent patterns and explore the meanings and processes associated with the interview data.

#### 2.1.4. Phase 4: Validation of the PDA

This will be carried out in order to assess the SDM outcomes of the PDA implementation in an experimental study in primary health centres compared with usual care, including an evaluation of the underlying process.

##### Participants

Clusters (primary health centres in the two Valencian Health Departments) will be randomised to use the PDA or usual care by using a computer-generated list based on the number of collaborating general practitioners. Based on a logistic regression, a cluster size of 12 clusters, each enrolling 30 patients (360 observations), will achieve 80% power (95% CI) to detect a mean difference of 0.35 between the groups in decisional conflict scores. Patients will be considered candidates for screening by their clinicians according to the European Society of Urology recommendations [23].

### 2.2. Data Collection

The study coordinator will enrol health care centres (clusters), and general practitioners will enrol patients. We will provide clinicians with one education session on the rationale for SDM in each centre. The intervention (PDA) will be applied at the cluster level to avoid contamination. Patients will be blind to the group designation. Factors influencing outcomes will be assessed at three levels: the health centre cluster, individual general practitioner and patient. The clinicians in the control group will proceed as in usual care. A research assistant blinded to the group assignment will collect information on patients’ demographic and clinical characteristics and on whether a PSA test was ordered from the medical records. A follow-up telephone survey of the patients will be conducted within 3 weeks of the visit in order to assess: (a) whether screening took place; (b) knowledge regarding prostate cancer (Knowledge of the Prostate Cancer tool [26]); (c) decisional conflict (ten-question Decisional Conflict Scale [27]) and (d) discussions regarding screening between patients and clinicians and the satisfaction with the screening decision (Satisfaction with Decision Scale [28]).

### 2.3. Statistical Analysis

Patients’ and general practitioners’ characteristics between clusters will be analysed using the *t* test or Wilcoxon rank sum test for the continuous variables and the χ^2^ test for the categorical variables. Modified Poisson regression with a sandwich estimator for standard errors will be used to estimate the relative risk, accounting for clustering by site using fixed effects.

### 2.4. Study Status

We are commencing the design of the knowledge surveys.

### 2.5. Ethics and Dissemination

The PROSHADE protocol was approved by the CEIC Sant Joan d’Alacant (20/041) on 8 January 2021. All participants will give oral/written informed consent prior to entry to the study by a member of the study team and will be made aware that participation is strictly voluntary. Participants may withdraw from the study at any time. The research adopts the principles of open science, and the findings will be published ensuring the participants’ confidentiality.

**Trial registration:** Clinical trials: NCT05187949 (https://clinicaltrials.gov/) (accessed on 1 July 2022).

## 3. Discussion

The complexity of decision making in oncology requires patients and clinicians to consider the benefits and risks of an increasing number of clinical options. In addition, patients and clinicians evaluate the options differently, and, therefore, they need new ways to evaluate all the relevant information and personal preferences to make an informed decision. SDM has been hailed as the key to successfully consider patients’ and clinicians’ preferences, and, consequently, researchers, clinicians, patients and health policy representatives have made a considerable effort to implement SDM in clinical practice [2]. Thus, this project will contribute to increasing the scientific knowledge on how to promote SDM in PC screening, often seen as a controversial decision in oncology.

Psychological factors have been negatively associated with preventive health behaviors such as cancer screenings. A previous systematic review aimed to evaluate the impact of cancer screening through patient-reported outcomes (PROs) assessment [29]. It showed that although the psychosocial impact of cancer screening is relatively low overall, even following false-positive test results, individuals with a higher risk of cancer tend to experience more symptoms of anxiety and distress during the screening process. Therefore, the results conclude that higher-risk individuals undergoing screening should be considered when carrying out prostate cancer screening.

In addition, the process carried out to develop this SDM strategy in PC screening could guide SDM research in other medical practice areas. There is little consensus on which PDA format is most effective, and it is a challenge to develop patient decision aids that can be routinely used in practice. In 2003, the International Patient Decision Aid Standards [24] established a consensus on quality standards between PDA developers. However, questions have been raised about its validity in practice. With this project, we will establish an empirical evaluation of this procedure, which is generically described, to develop a PDA. According to the available recommendations, we will assess patients’ and clinician’s decisional needs and knowledge to obtain a range of perspectives on the information that patients and clinicians need to have in order to have a maximal influence on SDM outcomes. Moreover, and given that gaps have been identified in presenting probabilistic PSA information to patients, we will contextualise relevant information for patients such as the likelihood of having a false-positive result, in order to help them in a decision according to their personal preferences. We will also systematically review the available literature about the barriers and facilitators of PC screening in primary care and develop an initial PDA prototype to be tested in the target population. Finally, we will assess the PDA implementation in clinical practice. To guarantee an adequate implementation in practice, we will measure key outcomes such as decisional conflict or patients’ satisfaction with the screening decision.

## 4. Challenges and Limitations

Study limitations include that the evaluation is focused on short-term outcomes, and we will not be able to assess the implementation of the PDA in long-term practice. However, to guarantee an adequate implementation in practice, we will measure key outcomes such as decisional conflict and patients’ satisfaction with the screening decision.

Patients will be involved in the conduct of this research. Promoting SDM in PC screening is not a straightforward task, and success will rely heavily on the interaction with the patients and healthcare workers during the entire research process. The inclusion of patients and clinicians in the design and development of the new decision aid will help foster ownership of the tool, which will help ensure a more sustained diffusion to clinical practice in the long term. The participation of scientific societies and the patient association ANCAP will also be crucial to this aim.

## 5. Conclusions

The well-designed PDA will provide balanced information on screening benefits and risks and help patients consider their personal preferences. This PDA will be able to improve the quality of the clinical interaction and communication between patients and clinicians, and patients could take a more active role in decision making. The promotion of SDM, when deciding on PSA screening, will have effects such as the development of collaborative deliberation between clinicians and patients, resulting in well-informed, empowered patients and preference-based decisions. Moreover, the promotion of SDM will result in safer, cost-effective and patient-aligned healthcare, including improvements in resource use and improved health outcomes.

## Figures and Tables

**Figure 1 ijerph-19-08904-f001:**
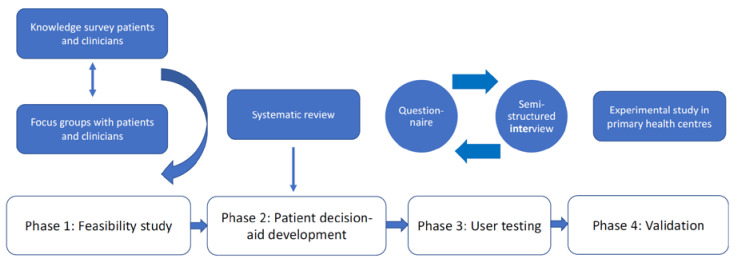
Description of the four phases included in the study.

## Data Availability

Not applicable.

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
