# Peer review of "PROSHADE Protocol: Designing and Evaluating a Decision Aid for Promoting Shared Decision Making in Opportunistic Screening for Prostate Cancer: A Mix-Method Study"

_ijerph, 2022, doi:10.3390/ijerph19158904_

Round 1

Reviewer 1 Report

This manuscript is of significance to a specific audience (clinicians and health care professionals). However, this is a tool that can be applied to other diseases, not only cancer, I believe that neurological disorders could also benefit with it.

Therefore, I accept this manuscript for publication in the International Journal of Environmental Research and Public Health, after the authors address only two small issues outlined below.

Minor issues:

-          Line 45: what does PDA means? Write here as it is in lines 64 and 65.

-          Lines 142, 150, 251: what does CI means?

Author Response

Response to Reviewer 1 Comments

This manuscript is of significance to a specific audience (clinicians and health care professionals). However, this is a tool that can be applied to other diseases, not only cancer, I believe that neurological disorders could also benefit with it.

Therefore, I accept this manuscript for publication in the International Journal of Environmental Research and Public Health, after the authors address only two small issues outlined below.

Minor issues:

-          Line 45: what does PDA means? Write here as it is in lines 64 and 65. 

PDA is Patient Decision Aid. We have included the meaning in line 45.

-          Lines 142, 150, 251: what does CI means?

CI is Confidence Interval. We have included the meaning in line 142.

Reviewer 2 Report

The protocol is interesting. I have few suggestions:

- please insert the Study Schedule figure in order to make the protocol easier to understand

- what about the psicological impact of prostate cancer screening? It may be interesting to consider. Please discuss it.

Author Response

Response to Reviewer 2 Comments

The protocol is interesting. I have few suggestions:

- please insert the Study Schedule figure in order to make the protocol easier to understand

In accordance with the reviewer’s advice, we have included a figure to show the different four phases of the study.

- what about the psicological impact of prostate cancer screening? It may be interesting to consider. Please discuss it.

We have included a paragraph in the discussion section dealing with this topic (page 8, lines 341-348). In addition, we have included a new reference (26)

Reviewer 3 Report

Lumbreras et al. have reported a protocol to promote a patient decision aid(PDA) to provide balanced information on prostate cancer (PCa) screening benefits and risks.

Some suggestions merit to be considered as follows.

Considering that the authors refer to PSA-based screening some more information about the limitation of the test should be considered. The sentence page 2 of 9 lines 54-58 is not correct. I suggest to consider this proposal"Current clinical practice guidelines for early detection of prostate cancer recommend for clinical decision-making a personalized prostate-specific antigen (PSA)-based management to improve the risk-benefit ratio of the screening strategy. Some important critical issues regarding the PSA determination (i.e.poor harmonization of the assays) in the clinical framework are, however, still neglected in current guidelines and a major focus of recommendations on those aspects would be needed to improve their effectiveness (references1,2).

After Lines 97-98 Some studies have proposed the use of well calibrated risk prediction model to define prostatespecific antigen (PSA) thresholds for identifying or excluding advanced prostate cancer (PCa) as an aid to personalize management of the diagnostic workup. The proposed PSA thresholds, being associated to the related predictive values, may allow an individualized approach to the diagnostic workup, assisting patients in making an informed decision (reference 3).

1) Ferraro S, Bussetti M, Rizzardi S, Braga F, and Panteghini M. Verification of Harmonization of Serum Total and Free Prostate-Specific Antigen (PSA) Measurements and Implications for Medical Decisions. Clin Chem. 2021;67:543-53. 2)Ferraro S, Bussetti M and Panteghini M. Serum prostate specific antigen (PSA) testing for early detection of prostate cancer: Managing the gap between clinical and laboratory practice.  Clin Chem 2021;67:602-609. 3)Ferraro, S.; Bussetti, M.;Bassani, N.; Rossi, R.S.; Incarbone,G.P.; Bianchi, F.; Maggioni, M.; Runza,L.; Ceriotti, F.; Panteghini, M. Definition of Outcome-Based Prostate-Specific Antigen (PSA) Thresholds for Advanced Prostate Cancer Risk Prediction. Cancers 2021, 13, 3381

Author Response

Response to Reviewer 3 Comments

Lumbreras et al. have reported a protocol to promote a patient decision aid(PDA) to provide balanced information on prostate cancer (PCa) screening benefits and risks.

Some suggestions merit to be considered as follows.

Considering that the authors refer to PSA-based screening some more information about the limitation of the test should be considered. The sentence page 2 of 9 lines 54-58 is not correct. I suggest considering this proposal "Current clinical practice guidelines for early detection of prostate cancer recommend for clinical decision-making a personalized prostate-specific antigen (PSA)-based management to improve the risk-benefit ratio of the screening strategy. Some important critical issues regarding the PSA determination (i.e.poor harmonization of the assays) in the clinical framework are, however, still neglected in current guidelines and a major focus of recommendations on those aspects would be needed to improve their effectiveness (references1,2).

Following the reviewer’s advice, we have incorporated this paragraph as well as the two references mentioned.

After Lines 97-98 Some studies have proposed the use of well calibrated risk prediction model to define prostate specific antigen (PSA) thresholds for identifying or excluding advanced prostate cancer (PCa) as an aid to personalize management of the diagnostic workup. The proposed PSA thresholds, being associated to the related predictive values, may allow an individualized approach to the diagnostic workup, assisting patients in making an informed decision (reference 3).

In accordance with the reviewer’s advice, we have incorporated this paragraph as well as the reference mentioned.

1) Ferraro S, Bussetti M, Rizzardi S, Braga F, and Panteghini M. Verification of Harmonization of Serum Total and Free Prostate-Specific Antigen (PSA) Measurements and Implications for Medical Decisions. Clin Chem. 2021;67:543-53.

2)Ferraro S, Bussetti M and Panteghini M. Serum prostate specific antigen (PSA) testing for early detection of prostate cancer: Managing the gap between clinical and laboratory practice.  Clin Chem 2021;67:602-609.

3)Ferraro, S.; Bussetti, M.;Bassani, N.; Rossi, R.S.; Incarbone,G.P.; Bianchi, F.; Maggioni, M.; Runza,L.; Ceriotti, F.; Panteghini, M. Definition of Outcome-Based Prostate-Specific Antigen (PSA) Thresholds for Advanced Prostate Cancer Risk Prediction. Cancers 2021, 13, 3381
